# Treatment Effects of Introducing the Neurosequential Model of Therapeutics in a Norwegian Residential Treatment Facility for Children Aged 7–13

**DOI:** 10.3390/children11050503

**Published:** 2024-04-23

**Authors:** Kaja Næss Johannessen, Ann-Karin Bakken, Erin P. Hambrick, Ole André Solbakken

**Affiliations:** 1Østbytunet Research and Development Unit, 1477 Fjellhamar, Norway; kaja@ostbytunet.no (K.N.J.); ann-karin@ostbytunet.no (A.-K.B.); 2Department of Psychology & Counseling, University of Missouri, Kansas City, MO 64110, USA; 3Department of Psychology, University of Oslo, 0313 Oslo, Norway; o.a.solbakken@psykologi.uio.no

**Keywords:** child and adolescent psychiatry, neurosequential model of therapeutics (NMT), individualized treatment plans, comorbid diagnoses, residential treatment, child behavior checklist (CBCL)

## Abstract

This study investigates the impact of the Neurosequential Model of Therapeutics (NMT) in child and adolescent psychiatric care, addressing a gap in current clinical methodologies that tend to focus on single problems rather than the interconnected nature of many real-life mental health issues. The study was conducted in a residential setting over an extended period, including children aged 7–13, to observe the effects of implementing NMT. The children presented with complex symptoms and multiple diagnoses. The methods incorporated the NMT approach, emphasizing individualized treatment plans based on each child’s unique brain development, and aimed at addressing multiple, interconnected problems simultaneously. Results from multilevel model analyses of behavioral difficulties, measured using the Child Behavior Checklist (CBCL), revealed substantial improvements in treatment effectiveness post-NMT implementation. Despite the limitations, such as a non-randomized participant selection and limited sample size, the findings strongly suggest that NMT enhances care effectiveness in real-world clinical settings, particularly for children with complex mental health issues. The study concludes that relationally oriented milieu therapy, and specifically the NMT approach, holds great promise for advancing pediatric psychiatric care, advocating for its broader application and further research to refine and substantiate its efficacy.

## 1. Introduction

Research on the treatment of children and adolescents demonstrates the considerable effects of psychotherapy [1]. A comprehensive list of Evidence Based Practices (EBP) addressing various clinical target problems is listed by Ng and colleagues [1]. However, studies on EBPs are most often not conducted in real-world clinical settings. Thus, these studies tend to be on patients with little comorbidity who are recruited rather than referred, and treatment is commonly delivered by research staff rather than real-world practitioners. When examining the effectiveness of these therapies in clinically representative settings, the magnitude of treatment effects appears to drop [2,3,4,5], a phenomenon Weisz and colleagues named the implementation cliff [6].

Weisz and colleagues [5,6] suggest a variety of reasons that might explain the difference between the effectiveness in controlled trials as compared to real-life clinical settings, e.g., greater presence of comorbid diagnoses in real-life clinical samples, time constraints on practitioners, and failure in organizations to support the introduction of new treatment approaches. Thus, it is essential to examine the effects of treatments under representative, real-world clinical conditions to identify those treatments that can optimally benefit clients who are treated in real-life mental health services.

The current research on EBTs in youth psychotherapy focuses mostly on single target problems only, e.g., conduct disorders, emotional disorders, ADHD, or OCD. However, a large proportion of patients referred to youth psychotherapy present with complex psychological difficulties and often comorbid diagnoses [7]. Accordingly, there is a need to fill the research gap between EBPs for single target problems and treatment for youth patients with complex mental health problems, given that these patients risk a range of social, emotional, and psychological challenges later in life if not provided with effective treatment [8,9]. In this study, we examine whether treatment in a naturalistic, real-world clinical setting for clients between the age of 7 and 13 presenting with complex psychiatric problems improve patient outcomes, and the effectiveness of a quality improvement effort, namely implementing the Neurosequential Model of Therapeutics (NMT), on patient outcomes. 

The Neurosequential Model of Therapeutics (NMT) is a relatively novel approach to the treatment of children with complex mental health problems, particularly those with histories of trauma and adversities [10,11,12]. Unlike traditional therapeutic models, NMT is not a specific therapeutic technique or intervention, but rather a neurodevelopmentally informed, biologically respectful framework designed to help the clinician understand the developmental and therapeutic needs of the patient and plan treatments accordingly, thereby enhancing the focus on individualized treatment interventions tailored to the patient’s needs [10,11,12]. The treatment model is transdiagnostic, focusing on the patient’s functioning in several domains, rather than planning the treatment based on diagnoses or single-target clinical problems.

The NMT process begins with a comprehensive review of the child’s history, including the chronological, relational, and developmental aspects, focusing on adverse childhood experiences in distinct developmental periods (prenatal, perinatal—0–2 months, infancy—3–12 months, early childhood—second year to age 4, childhood—age 4 through age 10 and youth—age 11 through age 18) and buffering relational support in the same developmental periods. This is followed by the completion of the NMT Metric, a structured assessment tool that helps to identify the child’s current neurodevelopmental status, including both strengths and vulnerabilities. The NMT Metric provides a visual representation of the child’s developmental trajectory, highlighting areas in which the child is functioning in line with their chronological age and areas in which their development may be delayed [10].

Based on this assessment, the NMT approach guides the development of individualized therapeutic plans. These plans are tailored to match the child’s current neurodevelopmental needs and are designed to help the child build or strengthen neural connections in areas that have been impacted by their experiences [10,11,12]. NMT draws upon principles of neuroplasticity as central elements in its theory of change, especially the principles of specificity (to change a network in the brain, that network needs to be used), repetition (frequency and repetition are needed to build new neural networks), and dosing and spacing of interventions (proper dosing of interventions and spacing in between is needed for efficient development of new neural networks).

Hence, treatment plans with specific interventions are made, focusing on what contribution each person in the patient’s “Therapeutic Web” (patient, family, school/childcare staff, therapists) is responsible for or needs for themselves, to be able to provide the necessary corrective experiences for the child. Hence, the model also focuses on real-life conditions that must be considered (like focusing on the needs of an exhausted parent or counter-therapeutic reactions towards a child among staff) if interventions are to be implemented properly [10,11,12]. Because the model puts so much emphasis on repetition and providing regular, small doses of therapeutic interventions rather than a once-a-week dosing, the model is a good fit for treatment to be delivered in residential settings.

NMT has shown promising effects in a variety of settings; juvenile justice and residential institutions [13,14], mental health clinics [15], foster care [16,17] and work with young children [18,19,20]. Being a young treatment model, there is, however, a need for further studies to establish its effectiveness. Importantly, to our knowledge, there has not been documented harm associated with the NMT, with one study even showing that NMT performed similarly to more established treatments as part of a randomized controlled trial [19].

### Aims of the Study

The aims and corresponding research questions of this study are two-fold.

Does residential treatment in a naturalistic, real-world clinical setting for clients between the age of 7 and 13 presenting with complex psychiatric problems improve patient outcomes?Are there differences in patient outcomes before and after the introduction of the Neurosequential Model of Therapeutics?

## 2. Materials and Methods

### 2.1. The Treatment Program and Its Components

The residential treatment center consists of three units with a total capacity of 24 children treated simultaneously. The center is closed on weekends and during vacations, thus each patient has a home base. On average, the patients spend two nights a week at the center, and the other days of the week, they return to their home base before bedtime. There is a school on the premises, solely for the patients receiving treatment at the treatment center.

The rhythm of each day is quite similar. Morning is spent in the treatment unit, then there is school until lunch, lunch is spent with staff at the treatment unit, and then back to school. After school, the rest of the day is spent at the treatment unit until bedtime, or the child goes home.

Treatment before implementation of NMT was relationally based, meaning that the main treatment focus was providing the children with enriching and corrective relational experiences (i.e., focusing on affect attunement and empathic responses) that grew their sense of self-worth and mastery. No particular EBP method was used consistently. Relational co-regulation was prevalent and elements from the Circle of Security and Marte Meo were frequently, but not systematically, used. Also, the children were frequently performing different activities with the staff, such as playing soccer, walking in the woods, biking, and general play inside or outdoors at the facility. These activities were not systematically planned according to a treatment plan, but more used as positive ways for the staff to spend time with the children.

Treatment after NMT implementation was still relationally focused, but with an added focus of the children’s developmental age in various functional domains, their stress level during the day, regulation through different means (relational and somatosensory), and their internal state and how this impacts the child’s capabilities and needs. The children’s developmental level in different domains was assessed using the NMT metric and specific interventions were planned to assist development in functional domains assessed to be underdeveloped. Typical behavior in each of the five states of ‘arousal’—calm, alert, alarm, fear, and terror (part of the NMT’s set of core heuristics)—was also assessed for each child, along with regulatory strategies that the child actively seeks in each state or that staff had seen could be beneficial for the child. More systematic exposure to somatosensory stimulation (e.g., walking, rocking, massage, use of fidgets) was introduced, both as part of a regular routine and as a therapeutic tool when the children’s stress response was highly activated. Somatosensory stimulation and predictability through added structure increased considerably at both the treatment unit and the school after the implementation of NMT, because of the staff’s increased understanding of the stress response system’s global effects on emotional, social, relational, behavioral, and cognitive functioning and what helps regulate the stress response.

### 2.2. Therapists and Training

Front line staff were originally trained in relationally oriented milieu therapy and consisted mostly of highly experienced therapists (bachelor’s in child protection and child welfare with additional clinical training, and clinically trained social workers). Each unit had a lead psychologist that was certified in the Neurosequential Model in Therapeutics and provided supervision within the Neurosequential framework to front line staff each morning during weekdays. The NMT Certification Process consists of a 90-h curriculum that integrates didactic learning, multimedia resources, and case-based training on topics such as recent trauma research, developmental psychology, neurobiology, and sociology to ensure a deep understanding of what a developmentally sensitive and trauma-informed approach could look like. Additionally, the program includes a maintenance training component and a biannual NMT Fidelity assessment, aimed at evaluating and ensuring high inter-rater reliability on the NMT Metrics, thereby maintaining the integrity and efficacy of the certification.

Front line staff were also exposed to the core concepts of the Neurosequential Model of Therapeutics through three full days of lectures for all staff once a year and one hour weekly in each unit. Each child’s progress and staff’s adherence to the model was discussed in a weekly meeting between frontline staff and the lead psychologist.

Equally, all teachers were exposed to the core concepts of the Neurosequential Model, taught by one of the psychologists certified in the model. Overall, they received 3 full-day trainings and 82.5 additional hours of training through the first year of implementation. There were also weekly discussions, concerning the pupils, between teachers and the lead psychologists trained in the model. Teachers engaged in this quality improvement initiative as part of their daily work.

The residential center began implementing the Neurosequential Model of Therapeutics (NMT) across the organization. More specifically, implementation of the NMT began in 2015, and most of the staff were trained in and started using the NMT (e.g., communicating with other staff about client care using NMT frameworks, treatment setting within the NMT framework, etc.) in 2016.

### 2.3. Procedures

The treatment center is part of the Norwegian national strategy for providing mental health treatment to the population. The center is owned by a private non-profit organization, and the treatment is funded by the government. Nationally, the mental health system is organized through the mental health clinics serving the population in designated surrounding catchment areas. Patients were referred to the treatment center by the out-patient clinics within the catchment area consisting of the counties surrounding Oslo. Patients referred are children with a complex presentation of symptoms and diagnoses, who to little or no extent have benefitted from prior out-patient treatment.

### 2.4. Participants

The participants were children aged 7–13 years receiving residential treatment. Data were collected between 2002 and 2020. All patients receiving residential treatment during this period were measured with the CBCL data obtained by caretakers, as part of evaluation of the treatment. Due to the beginning of implementation of NMT in 2015, data collected in 2015 were excluded from the study, including reports for clients who began services prior to 2015 but completed most of their data collection in 2015 (if clients had at least two timepoints collected before 2015, their data were retained; pre-2015 data were used for these clients). Criteria for inclusion in the study were enrollment in the residential treatment program within the defined study period and acceptance of the offer of treatment; there were no exclusion criteria applied.

The final sample included in the analyses comprised 69 clients (47 pre-NMT, 22 after NMT implementation). The unequal group size, though suboptimal, was a result of necessity, as the facility admits only a small number of patients every year and all receive long-term treatment (i.e., 3 years). Importantly, all possible and relevant cases at the facility from the beginning of the systematic data collection were included in the study, thus no bias in sample selection was present. In the sample, the mean age was 10.6 years (SD: 1.6, range: 7–13), 77% were male. Client data were recorded by clients’ caregivers assessing clients’ internalizing and externalizing symptoms at baseline (T1), after 1 year in treatment (T2), prior to client discharge (T3) and 1 year post treatment (T4).

### 2.5. Assessment Instruments

Symptoms were measured with the Child Behavior Checklist. The CBCL is a tool for measuring internalizing and externalizing symptoms in children and youth (aged 6–18) [21,22]. It is used to detect emotional and behavioral problems and competencies, and can be completed by the child’s caretaker, teacher, or by self-report. Only caretaker reports were used in this study. When completing the CBCL, the caretaker or support person indicates how true each of 112 problem behavior items is for the child. Outcomes are determined for significant problems regarding Internalizing Behavior (e.g., depression, anxiety), Externalizing Behavior (e.g., aggression, violence), and Total Problems. Numerous studies have demonstrated sound psychometric properties for the CBCL, and provided robust normative standards derived from extensive samples of both non-referred and referred children. Employment of standardized scores enables comparisons across genders and age cohorts [22].

### 2.6. Statistical Analyses

For analyzing the effectiveness of treatment delivered before and after implementation of NMT, we applied multilevel modeling with linear mixed models in the IBM SPSS, version 28.0. The use of multilevel modeling for the analysis of repeated measurements associated with the treatment courses of individual patients is strongly recommended in the field (see e.g., [23,24]). In such datasets, measurements are nested within individuals and measurements represent units at the first level, while individuals represent units at the second. A benefit of multilevel modeling is that variability in the number of assessments for different individuals is not in and of itself a problem [23]. This allows for variation in number of assessments during a time series, so that all cases assessed at least once contribute to calculation of the intercept, while all cases assessed more than once contribute to the calculation of the slope.

In our study, all 69 patients (47 pre-MNT, 22 after NMT implementation) contributed to the estimation of the intercept, while 63 (45 pre-NMT, 18 after NMT implementation) contributed to the estimation of the slopes of the treatment phase. There was thus little risk of bias in outcome estimates due to cases lost to drop-out or missing data in the treatment phase. In the follow-up phase, only 24 cases (17 pre-NMT, 7 after NMT implementation) delivered data, thus no comparison between treatment conditions was feasible in this phase and overall results for follow-up should be interpreted cautiously due to the high level of attrition. We also note that limitations of the data collection setting (i.e., data on very long-term treatment from a highly specialized in-patient facility admitting only a very small group of very vulnerable children every year) leads to both the relatively small overall sample and unequal group sizes. We used all available and relevant data over a period of 18 years but could not avoid the issue of unequal group sizes and a small n for the NMT group. However, the statistical procedures employed here, i.e., multilevel modelling, are known to be robust in the face of small samples and unbalanced designs, as has been noted by [25,26].

Multilevel modeling offers many ways of defining the passage of time and measurement occasions [24]. We are primarily interested in identifying the response to treatment before and after NMT implementation. Consequently, assessments are defined as fixed occasions and placed at constant distances across patients.

Preparatory data analyses. We completed a visual inspection of raw score and ordinary least square (OLS) plots to explore whether linear or nonlinear models will best fit the data, and whether most trajectories were best described by a one-piece or two-piece model [24]. Such inspection was performed for all dependent variables and a two-piece linear trajectory was determined to be best suited for most individual cases across outcomes with the first piece representing the treatment phase and the second piece representing follow-up.

Multilevel modeling. As noted, the multilevel models contained two levels of analysis representing repeated measurements nested within cases. Before further analyses, the time-varying dependent variables were centered so that intercepts were calculated at the time value of zero, removing problems with interpretation. Analyses examining change on outcome variables during treatment began by computing null models for each phase, which only contained the fixed effect of time, along with a random effect for the intercept and slope. As a second step, a fixed effect of treatment condition and its interaction with time was added (Model 1). This procedure estimates the magnitude of change on each outcome variable with a treatment condition as a moderator and tests the significance of those changes.

A significant treatment condition by time interaction in these multilevel models shows patients treated before and after NMT implementation have significantly different change trajectories. For the follow-up phase, we followed the same procedure but did not add a treatment condition as a moderator due to high levels of attrition and consequent inadequate statistical power. Due to the limited statistical power of the dataset and the proof-of-concept nature of the study, we employed one-tailed significance tests with a *p*-value set at 0.10 in all multilevel models in order to avoid a type II error. Because of the small overall sample and unequal groups, we conducted post hoc power analyses for the study in G*Power to test whether sample size and statistical power were adequate. Results of those analyses when applying the parameters specified above and testing for the smallest significant between-group effect size found in our data (d = 0.33) demonstrated a Critical F of 2.78 and a post hoc estimate of statistical power of 0.86. Sample size and power thus appears sufficient. Still, some caution is warranted in interpretation of the results, due to the small n in the NMT condition.

Effect sizes. For examining the magnitude of change, effect sizes (Cohen’s d) were calculated by dividing the estimated overall change scores by their corresponding standard deviations. To avoid inflating effects, estimated change scores were divided by the pooled standard deviations across all relevant measurement occasions. Cohen’s (1988) standards for evaluating magnitude of effect sizes were employed, with small effects classified as d = 0.2–0.5, medium effects as d = 0.5–0.8, and large effects as d ≥ 0.8.

## 3. Results

### 3.1. Sample Characteristics

The distribution of diagnoses (categorized using ICD main categories) were as follows: 42% of the patients met the criteria for hyperkinetic disorders; 17% for conduct disorders, 17% met the criteria for reactions to severe stress and adjustment disorders; 13% for tic disorders; 12% for emotional disorders; and 9% for pervasive developmental disorders. Less commonly observed diagnoses, identified in a range of 1% to 6% of the patients, encompassed dissociative disorders, mixed conduct and emotional disorders, attachment disorders, enuresis and encopresis, various other anxiety disorders, speech and language-related disorders, and unspecified disorders of psychological development. While 2 patients did not have any diagnoses, 21 patients had 1 diagnosis, 18 patients had 2 diagnoses, and 16 patients had 3 or more diagnoses.

Patient flow and data completeness. Figure 1 shows the flow of clients from enrolment to follow-up in the two treatment conditions. All 69 clients entering treatment delivered the baseline data and completed the treatment program. A total of 63 clients delivered data mid-treatment, while 41 did so at termination. A total of 24 clients delivered follow-up data (17 in the pre-NMT condition, 7 in the NMT condition). Forty-five clients were lost to follow-up (30 in the pre-NMT condition, 15 in the NMT condition). Consequently, results for follow-up analyses must be regarded as tentative.

### 3.2. Effectiveness of Treatment before and after NMT Implementation

Results of multilevel analyses across the three outcome variables for clients before and after NMT implementation during the treatment phase are presented in Table 1. The analyses demonstrated statistically significant improvements for clients on treatment in all three outcome measures. In addition, there were statistically significant interactions between time and treatment condition (before vs. after NMT implementation) for two of the three outcome variables (CBCL-Total and Internalizing), indicating that for both variables, NMT was statistically superior to non-NMT.

Results from multilevel models of the follow-up phase after the end of treatment demonstrated the stability of the scores and accordingly, no significant changes in any of the outcome variables.

### 3.3. Effect Sizes

To make the results comparable with other studies and across outcome variables, effect sizes (Cohen’s d) were computed. Figure 2 displays the obtained effect sizes from the beginning of treatment to termination for each outcome variable for clients receiving treatment before and after NMT implementation. On the CBCL-Total, the effect size was 0.29 before and 0.72 after NMT implementation. On Externalizing, it was 0.35 before and 0.70 after NMT implementation, and on Internalizing, the effect size was 0.20 before and 0.53 after NMT implementation. Thus, interestingly, all effects were small to moderate pre-NMT and moderate to large with NMT.

## 4. Discussion

This study examined two separate yet connected research questions. First, we wanted to explore the effectiveness of residential treatment in a naturalistic, real-world clinical setting for clients with complex mental health problems, aged 7 to 13. Second, we wanted to see if the implementation of the Neurosequential Model of Therapeutics altered treatment effectiveness. This research question was considered to be important given that many times, clinical outcomes in real-world settings are not realized, even when using evidence-informed approaches [6].

### 4.1. Summary of Main Findings

Analyses indicated the significant effects of treatment, suggesting that relationally oriented milieu therapy in a residential treatment setting may be effective in treating children presenting with complex psychiatric problems. The analyses of effect sizes (Cohen’s d) indicated that the effects of treatment delivered before the implementation of NMT were small to moderate. Effect sizes after implementation of NMT, on the other hand, were moderate to large on all three outcome variables (Externalizing, Internalizing, and Total problem score), suggesting that the implementation of the Neurosequential Model of Therapeutics may have improved the effectiveness of the treatment delivered.

### 4.2. Why NMT May Perform Better?

The fact that the Neurosequential Model of Therapeutics (NMT) demonstrated increased effectiveness in the study may potentially be attributed to several distinct factors: First, the transdiagnostic treatment aspect of NMT. Patients in the study were diagnosed with multiple conditions concurrently. While many Evidence-Based Practices (EBPs) are tailored to address single problems, NMT takes on a more encompassing approach. This broader perspective allows for the treatment of multiple issues at once, more aptly reflecting real-life clinical scenarios. Second, the individualized treatment approach of NMT stands out. It does not just offer generic solutions; instead, it focuses on understanding each patient’s unique neurodevelopmental profile, strengths, and challenges. Assisted by the NMT Metric, clinicians can better pinpoint and act upon each patient’s specific needs. By ensuring that the interventions align closely with the child’s unique requirements, NMT elevates the potential for positive therapeutic outcomes and a faster return to a healthy developmental trajectory. After the implementation of NMT, the milieu therapy shifted from being focused on providing a general friendly, relational, and developmentally focused environment that were thought to be helpful for all children, to a more tailored treatment approach towards each child, based on the NMT Metric. Lead psychologist and staff became better at recognizing developmental delays in each child and systematically offering experiences to bridge these gaps.

The wide-ranging therapeutic interventions of NMT may also play a pivotal role. Recognizing that multiple environmental factors influence a child’s well-being, NMT emphasizes interventions not just for the child, but also across their broader environment, including family, school, and community contexts. This multifaceted approach spans from specialized therapies, like EMDR, to broader interventions, such as in-school support and community education. Importantly, NMT’s treatment strategies are designed with real-world practicalities in mind, promoting adherence and implementation. After implementing NMT, both front line staff and lead psychologist became more focused on providing the caregivers with psychoeducational knowledge about development and brain functioning as related to behavior and symptoms and explaining the rationale behind activities chosen for their children. Encouragement of the caregivers to provide the same experiences at home increased and both front line staff and lead psychologist explained that they felt this part of the work became easier after being introduced to the Neurosequential Model theoretical framework.

Finally, there is the foundation of neuroplasticity principles. NMT is built upon the understanding that therapeutic success involves nurturing new neural connections in the brain. By embracing principles like specificity, repetition, and dosing, NMT harnesses the brain’s inherent ability to adapt, thereby optimizing the therapeutic process.

### 4.3. Clinical Implications

There are several key takeaways from this study for professionals in pediatric psychiatric care. As mentioned in the introduction, there is an existing gap in today’s clinical methods. Most current treatments focus on just one problem at a time. However, many young patients face multiple interconnected issues, and we need treatments that can handle this complexity [27]. The results of this study suggest that NMT might be one such solution. We found better results with NMT than with previous intervention systems employed, suggesting that it not only has the potential to bridge this research gap, but could also help to enhance the effectiveness of care in residential settings.

One of NMT’s main strengths may be its focus on individualized treatment. By using information about a child’s brain development, NMT provides treatments that match the specific needs of each patient. This is especially important for children with complex mental health problems. Additionally, the NMT treatment planning process allows for focusing on multiple, interconnected problems at once. This might make the treatment process simpler and more direct for both patients and their families. Furthermore, on a practical level, NMT is flexible. It offers strategies that work well in different settings, making it a useful tool in real-life settings. This study suggests that treatment methods that allow for flexibility, target several therapeutic problems at once, and aids in the development of individually tailored treatment plans based on current neurodevelopmental status might be more beneficial when treating children with complex mental health issues in real-life clinical settings.

Indeed, key takeaways for those seeking clinical program improvements might be that an individualized assessment, and trauma-informed and relationally rich intervention, when implemented in a robust manner and utilizing all staff (teachers, clinicians, administration), may lead to noticeable outcomes in real-world settings, even with complex patients. Highly individualized assessments and treatments have been identified as useful with other highly trauma-exposed youth [28]. Indeed, while scaling up teachers and clinicians may have been time-consuming, results suggest that the investment may have been worthwhile.

### 4.4. Study Strengths and Limitations

The research scrutinized NMT’s application in a realistic clinical environment over an extended period, offering insights into its lasting effects. Additionally, by encompassing a diverse age group, the study’s findings could apply to children across various developmental stages.

There are also a few limitations to this naturalistic study that must be noted. Firstly, the participants were not randomized, which means that the study may not be entirely representative of the population being studied. Additionally, there was not a control group, meaning that causation cannot be inferred. Secondly, there may be time effects that could influence the results, which is another possible limitation. Another limitation is related to the fact that there was no formal evaluation of the therapist’s adherence to or competence in the Neurosequential Model. Therefore, we cannot be entirely certain that the treatment we have measured is in line with the principles of the model. Additionally, the study’s fairly limited sample size of 69 may affect the robustness of its results.

There were also some attritions during follow-up that could bias interpretations of the long-term outcomes. Finally, sole dependence on the Child Behavior Checklist might limit the study’s breadth, and a diversified assessment approach would offer a more well-rounded perspective on treatment impact.

### 4.5. Recommendations for Future Research

Future research should broaden the sample size and diversity to strengthen our confidence in the validity of conclusions about NMT’s effects in pediatric settings. In addition, there should be a focus on employing multiple assessment tools, including qualitative techniques, to mediate the limitations of this study and further research the potential of NMT. Further, studies should also disentangle NMT’s individual treatment components to determine the most impactful ones, aiding clinicians in refining treatment plans.

## 5. Conclusions

The present study demonstrated the effectiveness of treatment in a clinically representative, real-world institution delivering residential treatment. Furthermore, the study showed significant and substantial improvement in the effectiveness of treatment after introduction of the Neurosequential Model of Therapeutics (NMT). Together, the results suggest that relationally oriented milieu therapy is beneficial for children with complex psychiatric problems and that the Neurosequential Model in Therapeutics is a highly promising treatment model that may further increase treatment effects in real-world settings.

## Figures and Tables

**Figure 1 children-11-00503-f001:**
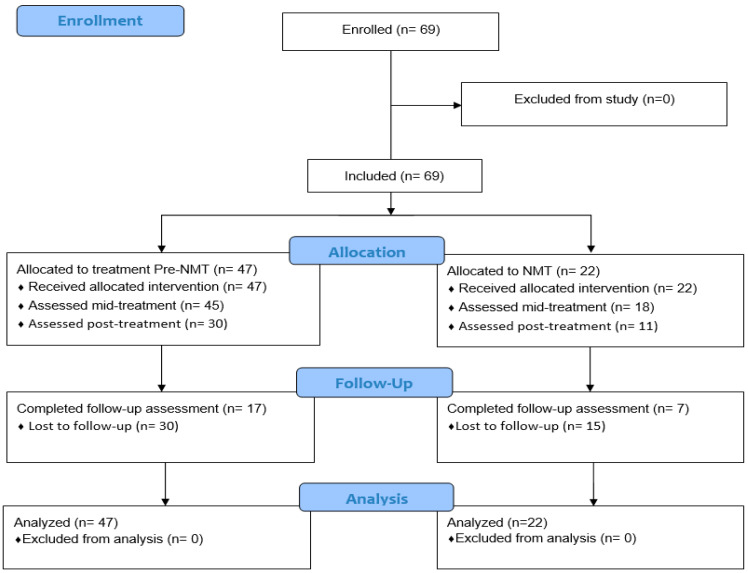
Patient flow and data completeness.

**Figure 2 children-11-00503-f002:**
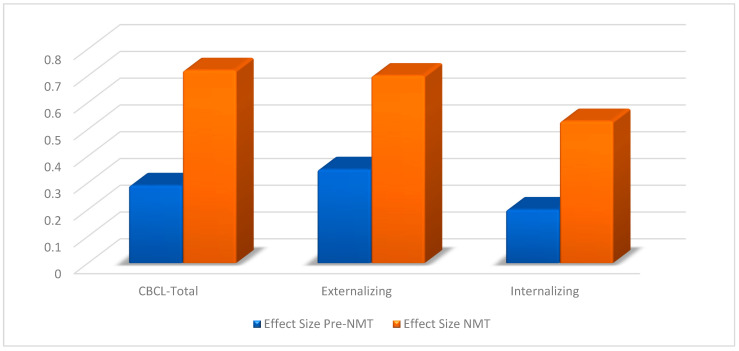
Effect sizes for patients treated at the center pre- and post-NMT implementation. Note: CBCL-Total = overall mean score on the CBCL. Externalizing = mean score for externalizing problems on the CBCL. Internalizing = mean score for internalizing problems on the CBCL. Effect Size = Cohen’s d. Pre-NMT = treatment delivered before the implementation of NMT. NMT = treatment delivered after the implementation of NMT.

**Table 1 children-11-00503-t001:** Results of the multilevel model analyses of the behavioral difficulties during treatment before and after NMT implementation.

	CBCL Total Score			Externalizing			Internalizing		
	Model 0	Model 1		Model 0	Model 1		Model 0	Model 1	
Fixed effects	Est	Est	*t* (*df*)	Est	Est	*t* (*df*)	Est	Est	*t* (*df*)
Intercept	72.20 (3.15)	72.06 (3.91)	18.44 (87.11)	25.53 (1.28)	26.10 (1.58)	16.56 (82.69)	16.36 (1.05)	16.39 (1.31)	12.55 (86.69)
Time	−5.73 (1.67)	−4.24 (1.97)	−2.15 (34.81)	−2.49 (0.75)	−2.03 (0.87)	−2.33 (46.01)	−1.35 (0.50)	−0.97 * (0.59)	−1.65 (42.04)
NMT	-	1.30 (6.77)	0.19 (91.92)	-	−1.41 (2.74)	−0.51 (87.69)	-	0.15 (2.25)	0.07 (90.77)
Time × NMT	-	−6.25 * (4.04)	−1.55 (47.07)	-	−2.00 (1.77)	−1.13 (60.55)	-	−1.57 * (1.21)	1.31 (55.64)
Parameters	Est	Est	Wald Z	Est	Est	Wald Z	Est	Est	Wald Z
Residual	218.25 (57.22)	201.55 (56.01)	3.60	42.89 (10.82)	40.95 (10.61)	3.86	15.86 (4.01)	15.14 (2.89)	3.89
Variance in intercept	620.27 (139.16)	654.02 (143.08)	4.57	92.71 (23.60)	96.79 (24.23)	3.99	79.06 (15.23)	81.43 (15.59)	5.22
Variance in slopes	3.54 (53.26)	21.50 (56.88)	−0.38	1.69 (9.29)	3.57 (9.47)	−0.38	2.42 (4.05)	3.52 (4.19)	−1.11
Akaike Information Criterion	1585.38	1572.74	-	1289.03	1279.74	-	1186.21	1178.84	-

Note. Standard errors are in parenthesis. Degrees of freedom (*df*) and *t*-values are given only for the final model. Estimations were performed by the method of restricted maximum likelihood (REML). * *p* < 0.10; *p* < 0.05.

## Data Availability

The data presented in this study are available on request from the corresponding author. Data are not publicly available due to privacy reasons.

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
