# Peer review of "Treatment Effects of Introducing the Neurosequential Model of Therapeutics in a Norwegian Residential Treatment Facility for Children Aged 7–13"

_children, 2024, doi:10.3390/children11050503_

Round 1

Reviewer 1 Report

Comments and Suggestions for Authors

Dear authors, I appreciate the opportunity to have been able to review your work.

Next I would like to send you some comments:

- I consider that the introduction is well written and consistent with the topic covered.

- uniting the research question with the objective raises a lot of doubts in me. The objective should be clearly described but is not listed as such in the document.

- When they indicate that the minors received "enriching relational experiences", what do they mean?

- What type of therapists participated? I think it is not clear enough.

- Did the teachers who participated and received training do so voluntarily and altruistically or was it part of their daily work?

- The work has good results and good applicability. They have detailed its strengths and limitations excellently.

- What do you think should be done so that it could be extrapolated to the rest of the population, to other towns, cities or even countries?

Thank you

Author Response

- I consider that the introduction is well written and consistent with the topic covered.

            We appreciate this affirmation.

- uniting the research question with the objective raises a lot of doubts in me. The objective should be clearly described but is not listed as such in the document.

We have now clearly stated the objective on lines 54-58: In this study, we examine whether treatment in a naturalistic, real-world clinical setting for clients between the age of 7 and 13 presenting with complex psychiatric problems improve patient outcomes and also the effectiveness of a quality improvement effort, namely implementing the Neurosequential Model of Therapeutics (NMT), on patient outcomes.

- When they indicate that the minors received "enriching relational experiences", what do they mean?

Thank you for pointing out the lack of clarity, we agree that this was unclear in the text. We have added a sentence defining this concept in line 123.

- What type of therapists participated? I think it is not clear enough.

We agree that information given was insufficient and have now clarified information about the background of therapists in line 150-151.

- Did the teachers who participated and received training do so voluntarily and altruistically or was it part of their daily work?

            The teachers participating in the study did so as part of their daily work (added to line 170).

- The work has good results and good applicability. They have detailed its strengths and limitations excellently.

            Thank you!

- What do you think should be done so that it could be extrapolated to the rest of the population, to other towns, cities or even countries?

We added the following to the text, taking care not to “speak beyond” the current data: Key take-aways for those seeking clinical program improvements might be that an individualized, assessment and trauma-informed, and relationally rich intervention, when implemented in a robust manner and utilizing all staff (teachers, clinicians, administration), can lead to noticeable outcomes in real-world settings, even with complex patients. Indeed, while scaling up teachers and clinicians may have been time-consuming, results suggest that the investment may have been worthwhile.

Reviewer 2 Report

Comments and Suggestions for Authors

In this study designed to investigate the effect of the Neurosequence Therapeutic Model, striking results were achieved.Long-term follow-up is a practice that makes the study powerful. However, I see that there is no normal sample distribution over the years. This situation may negatively affect the robustness of the study. In addition, it is seen that there is a difference between the control group and the experimental group and this difference is large in number. These may cause problems for operation. I also have to say with regret that the discussion was written very briefly and was inadequate. If you can demonstrate the adequacy of your sample by making a calculation for the sample and support and interpret your discussion with stronger sources, I can reconsider the study.

Comments on the Quality of English Language

In this study designed to investigate the effect of the Neurosequence Therapeutic Model, striking results were achieved.Long-term follow-up is a practice that makes the study powerful. However, I see that there is no normal sample distribution over the years. This situation may negatively affect the robustness of the study. In addition, it is seen that there is a difference between the control group and the experimental group and this difference is large in number. These may cause problems for operation. I also have to say with regret that the discussion was written very briefly and was inadequate. If you can demonstrate the adequacy of your sample by making a calculation for the sample and support and interpret your discussion with stronger sources, I can reconsider the study.

Author Response

-In this study designed to investigate the effect of the Neurosequence Therapeutic Model, striking results were achieved.

            Thank you!

Long-term follow-up is a practice that makes the study powerful. However, I see that there is no normal sample distribution over the years. This situation may negatively affect the robustness of the study.

            -First, we reduced causal language in the Discussion to account for the lack of a control group, and added this statement: Additionally, there was not a control group, meaning that causation cannot be inferred.

In addition, it is seen that there is a difference between the control group and the experimental group and this difference is large in number. These may cause problems for operation.

-Yes, thank you for pointing this out. Indeed, limitations of the data collection setting (i.e., data on very long-term treatment from a highly specialized in-patient facility admitting only a very small group of very vulnerable children every year) leads to both a relatively small sample size and the unbalanced group sizes you refer to. We have used all available data over a period of 18 years, but as you note, still cannot avoid the issue of unequal group sizes and a small n for one group. Luckily, there are statistical procedures, such as multilevel modelling that are robust in the face of such unbalanced designs and we have applied them. As has been noted by Snijders (2005), in multilevel modelling (as applied in our paper) the total number of level 1 observations (in our case the number of assessments of each child across time) is the strongest determinant of statistical precision (and hence power), while the number of observations at level 2 (in our case the total number of children, i.e., 69) are less important for the power and robustness of such tests. This implies that the sample size at the highest level is the main limiting characteristic of the design (and in our case this number is 197). As you no doubt know, with the traditional repeated measures ANOVA, an unbalanced design often leads to problems with the analyses, however, this is not the case in multilevel modeling. In line with this, another of the MLM-pioneers, Joop Hox (along with Maas; 2005) has shown that level 2 samples of 50 or less may lead to biased estimates, while being reasonably unbiased above that number.

References to support statistical choices:

Maas, C. J., & Hox, J. J. (2005). Sufficient sample sizes for multilevel modeling. Methodology, 1(3), 86-92.

Snijders, Tom A.B. ‘Power and Sample Size in Multilevel Linear Models’. In: B.S. Everitt and D.C. Howell (eds.), Encyclopedia of Statistics in Behavioral Science. Volume 3, 1570–1573. Chicester (etc.): Wiley, 2005.

I also have to say with regret that the discussion was written very briefly and was inadequate. If you can demonstrate the adequacy of your sample by making a calculation for the sample (Ole)?

-As noted above, we are confident that the inherent problems with the unbalanced sample have been handled adequately by the multilevel modelling techniques applied. However, the issue of the low number of participants in one of the groups remains. Of course, the main problem with underpowered studies is the risk of not detecting effects of actual importance. Accordingly, we adjusted the threshold for rejecting a difference between groups by applying one-tailed tests of significance and a p-value set at 0.10 order to avoid type II error (see lines 258-259). For your convenience, we have also conducted post hoc power analyses, to test the adequacy of the sample given these parameters in G*Power. Results of those analyses when entering the smallest significant between-group effect size in our data (d = .33) yielded a Critical F of 2.78 and a post hoc estimate of statistical power of 0.86. Hopefully, this eases your concerns.    

and support and interpret your discussion with stronger sources, I can reconsider the study.

We respectfully disagree that the Discussion is wholly inadequate, as it summarizes findings, and clearly states limitations and conclusions without speaking beyond the current data. That said, to be responsive, we have worked to buff up the Discussion by responding to other reviewers’ concerns, and also by doing the following:

Reducing causal language

            Expanding on potential limitations

Restating the objective of the paper at the outset (This research question was considered to be important given that many times, clinical outcomes in real-world settings are not realized, even when using evidence-informed approaches [6].)

Adding in reputable references to support claims made.

Round 2

Reviewer 2 Report

Comments and Suggestions for Authors

This study investigates the impact of the Neurosequential Model of Therapeutics (NMP). However, there are shortcomings. For the introduction part, he mentions NMP. It explains why it is made. However, there is no information about whether it is harmful to children. It would be good to add something on this subject. Yes, it is a beneficial and new initiative for children's mental health, but what harm can it do to the child? You need to know these. In addition, the complexity of the method part, the lack of sample selection criteria, the inconsistency of the number of samples between the groups and the large difference between them are quite thought-provoking. Please explain with supporting sentences the method by which you selected the sample and why there is such a large difference between the groups. Maybe there may be studies done in this way. You can calculate the sample with the standard deviations in these studies and add it to your method section. Then this problem will disappear. I think sample calculation is a very important criterion. You can easily make this calculation from Gpower. It will be enough to look at the standard deviations of the averages of similar studies.

Comments on the Quality of English Language

Minor editing of English language required

Author Response

Comment 1: This study investigates the impact of the Neurosequential
Model of Therapeutics (NMP). However, there are shortcomings. For the
introduction part, he mentions NMP. It explains why it is made. However,
there is no information about whether it is harmful to children. It
would be good to add something on this subject. Yes, it is a beneficial
and new initiative for children's mental health, but what harm can it do
to the child? You need to know these.

Reply: We have added reflections and statements on what is known about
possible harmful effects of NMT that we have found in the literature at the end of the Introduction. We are not aware of any published harms, including the outcomes of a randomized controlled trial using the model. 

Comment 2: In addition, the complexity of the method part, the lack of
sample selection criteria, the inconsistency of the number of samples
between the groups and the large difference between them are quite
thought-provoking. Please explain with supporting sentences the method
by which you selected the sample and why there is such a large
difference between the groups. Maybe there may be studies done in this
way. You can calculate the sample with the standard deviations in these
studies and add it to your method section. Then this problem will
disappear. I think sample calculation is a very important criterion. You
can easily make this calculation from Gpower. It will be enough to look
at the standard deviations of the averages of similar studies.

Reply: The methods section has been amended with more detailed
information on sample selection and why the two groups are unequal.
Also, we have added more justification for the analyses, added more
caution as to interpretation, and included descriptions of post hoc
power analyses testing the adequacy of the sample based on the
statistical parameters outlined for the study in G*Power. Results of
those analyses when testing for the smallest significant between-group
effect size in our data (d = .33) yielded a Critical F of 2.78 and a
post hoc estimate of statistical power of 0.86. Finally, all conclusions
have been critically revised and made more tentative wherever deemed
necessary. Hopefully, this satisfies what you indicated was needed. We
were not totally clear on what you meant by calculating the sample with
the standard deviations in other studies (Re you comment: “Maybe there
may be studies done in this way. You can calculate the sample with the
standard deviations in these studies and add it to your method
section”). Thus, if you want us to do the power analyses differently,
just specify what you think would be a reasonable procedure and we will
provide the relevant analyses.

Round 3

Reviewer 2 Report

Comments and Suggestions for Authors

The authors have done their best to make the requested corrections.